# Development and Characterization of Films with Propolis to Inhibit Mold Contamination in the Dairy Industry

**DOI:** 10.3390/foods12081633

**Published:** 2023-04-13

**Authors:** Romina L. Abarca, Francisco Vargas, Javiera Medina, Juan Carlos Paredes, Bernardo Carrillo López, Pablo A. Ortiz, Einar Vargas-Bello-Pérez

**Affiliations:** 1Departamento de Ciencias Animales, Facultad de Agronomía e Ingeniería Forestal, Pontificia Universidad Católica de Chile, Macul, Santiago 7820436, Chile; 2Instituto de Ciencia y Tecnología de los Alimentos, Facultad de Ciencias Agrarias, Universidad Austral, Avda. Julio Sarrazín s/n, Isla Teja, Valdivia 5090000, Chile; 3Instituto de Química, Facultad de Ciencia, Universidad Austral de Chile, Isla Teja, Valdivia 5090000, Chile; 4Núcleo de Química y Bioquímica, Facultad de Estudios Interdisciplinarios, Universidad Mayor, Santiago 8580745, Chile; 5Department of Animal Sciences, School of Agriculture, Policy and Development, University of Reading, P.O. Box 237, Earley Gate, Reading RG6 6EU, UK; 6Facultad de Zootecnia y Ecología, Universidad Autónoma de Chihuahua, Periférico R. Aldama Km 1, Chihuahua 31031, Mexico

**Keywords:** sodium alginate, propolis, fungi, antifungal activity, ripened cheese, active film, active coating

## Abstract

Due to the number of polyphenols with multiple biological activities, propolis has high potential to be used as an active agent in food protective films. Therefore, this study aimed to develop and characterize a sodium alginate film with ethanolic extract of propolis (EEP) for its potential use as protective active packaging against filamentous fungi in ripened cheese. Three different concentrations of EEP were analyzed: 0, 5 and 10% *w*/*v*. The films obtained were characterized, assessing thermal and physicochemical properties, as well as the concentration of polyphenols in the EEP and antifungal activity of the active films. The incorporation of EEP in the films generated thermal stability with respect to the loss of mass. Total color values (ΔE) of the films were affected by the incorporation of the different concentrations of EEP, showing a decrease in luminosity (L*) of the films, while the chromatic parameters a* and b* increased in direct proportion to the EEP concentration. Antifungal activity was observed with a fungistatic mode of action, stopping the growth of the fungus in cheeses without development of filamentous molds, thus increasing the shelf life of the ripened cheese under the analytical conditions, over 30 days at room temperature. Overall, EEP can be used to prevent growth and proliferation of spoilage microorganisms in cheese.

## 1. Introduction

Food spoilage is a major problem for the food industry and consumers, as it makes products unacceptable for consumption and consequently generates a significant economic loss [1]. The environment of food production and processing is usually one of the main sources of contamination, the air being an important disseminator of fungal spores [2]. Fungi are one of the main spoilage organisms in dairy products, specifically in fermented dairy products and low-moisture products such as ripened cheese [3].

The main fungi involved in the spoilage of these dairy products are yeasts of the genus *Candida*, *Galactomyces* and *Yarrowia*, and molds of the genus *Penicillium*, *Alternaria*, *Fusarium*, *Mucor*, *Talaromyces* and *Cladosporium* [4,5,6,7]. These organisms produce enzymes that break down lipids, proteins, and carbohydrates, leading to a variety of undesirable sensory qualities [8].

Fungal contamination of dairy foods can occur at different stages, whether in milking parlors, in processing facilities, or in consumers’ homes. Fungi growth during milk reception is minimal, due to the excess of psychotropic bacteria such as *Pseudomonas* spp. In addition, fungi, molds, and yeasts are not resistant to heat treatments, so the sources of deterioration are established in the manufacturing and storage process, mainly due to fungal spores that circulate in the environment of cheese plants [5]. Cheese ripening rooms provide an excellent environment for the growth of these undesirable molds [9]. This is due to the extrinsic and intrinsic conditions favorable for the development of molds and the other microorganisms mentioned previously, such as temperature, relative humidity, cheese maturation times, activity water (aw) and their nutritional composition [10,11]. Different traditional methods are implemented to control fungi, including air filtration systems, cleaning and disinfection procedures, heat treatment, refrigeration, modified atmosphere packaging and the use of preservative chemicals that are considered as food additives [5]. An example of these chemicals is natamycin, which exerts an antifungal effect directly on the food matrix [12].

Innovations are constantly being generated in food packaging, always with the aim of creating a more efficient quality of preservation system that, at the same time, improves the attractiveness and marketability of food. Currently, the development of food active packaging has focused on bio-based functional materials that incorporate compounds and natural active ingredients, such as essential oils, botanical extracts and propolis, standing out for its high capacity to inhibit the growth and proliferation of various microorganisms [13,14,15,16]. Within these materials is sodium alginate, a salt of alginic acid that is isolated from brown algae and can be synthesized by microorganisms. It is considered an inexpensive, non-toxic, biodegradable, and biocompatible biopolymer that is safely used in food as a thickener, gelling agent, and emulsion stabilizer [17,18]. On the other hand, propolis is a resinous mixture of substances collected and used by bees to protect the entrance of the hive. It is mainly composed of resins and vegetable balsam, wax, essential and aromatic oils, pollen and other substances. The main function of propolis is to act as an antibacterial and antifungal agent [19,20]. The most active components of propolis are polyphenols (flavonoids, phenolic acids and their esters), steroids, terpenoids, amino acids, ketones and volatile aldehydes [21,22,23,24].

Due to its antifungal properties, propolis is considered as a functional compound in food packaging [25]. Due to the number of polyphenols with multiple biological activities, propolis has high potential to be used as an active agent in food protective films. Therefore, this study aimed to develop and characterize a sodium alginate film with ethanolic extract of propolis (EEP) for its potential use as protective active packaging against filamentous fungi in ripened cheese thus replacing antifungal compounds of a chemical nature currently used by the cheese industry. 

When generating films and/or coatings with the incorporation of active agents (such as EEP), it is highly relevant, before evaluating changes in the sensory characteristics of the food to be coated, that determination of the properties of the material and/or coating system must be characterized/evaluated. These characteristics are related to microbiological, thermal, physical and chemical properties. Based on the foregoing, this study performed a set of analyzes of the coatings made and a preliminary analysis of application to a dairy product (cheese), with a projection to carry out subsequent studies such as evaluation of sensory characteristics and shelf life.

## 2. Materials and Methods

The films/coatings were prepared using sodium alginate (QUIMSA, Productos Químicos y Laboratorio, Santiago, Chile), distilled water as solvent and glycerol as plasticizer (Winkler Ltda, Santiago, Chile). Propolis was used as the antifungal agent, obtained from San José de la Mariquina, Valdivia (South of Chile). The filamentous fungi were obtained from the environmental analysis of cheese ripening chambers and were cultured in 9-cm Petri dishes with potato dextrose agar (PDA). Artisanal cheese was in its fresh and ripened state. The cheeses were manufactured by local producers (Los Ríos, Chile) with pasteurized natural milk. 

The material was obtained by means of the casting technique in the form of a film to carry out its characterization and then, as a preliminary analysis, it was applied as a coating on pieces of fresh and ripened cheese. Cheese pieces were preserved at 12 °C and 85% relative humidity. Three different concentrations of EEP were applied to the cheese pieces at 5 and 10% *w*/*v*. The control had no EEP and all treatments were stored under the same conditions.

### 2.1. Antifungal Agent Extraction

Propolis was extracted with the Soxhlet method described by de Lima et al. (2016), [26] with slight modifications. As a solvent, 99.9% methanol was used to obtain a total extract of propolis. From this total extract, different concentrations were formulated with 99.9% ethanol to obtain EEP.

### 2.2. Determination of Total Polyphenols

Polyphenol total concentration was determined by colorimetry with the Folin-Ciocalteu assay. This method works by reducing polyphenols in an alkaline solution, resulting in the formation of a blue coloring complex. As indicated, [27], the method consisted of taking 40 μL of extract, to which 3.16 mL of distilled water, 200 μL of Folin-Ciocalteu reagent and 600 µL of 20% sodium carbonate penta-hydratate (Na_2_CO_3_·5H_2_O) were added. The mixture was stirred and left in the dark for two hours. Then, the mixture was read in a spectrophotometer at a wavelength of 765 nm. To obtain the concentration of polyphenols, a calibration curve was carried out using gallic acid as the standard, with measuring absorbance at 765 nm in triplicate. Results were expressed as gallic acid equivalent (GAE).

### 2.3. Minimum Inhibitory Concentration (MIC)

The minimum inhibitory concentration analysis was performed under the protocol described [28], (Abarca et al., 2017), with minor modifications. To determine the MIC, strains present in the cheese ripening chambers were isolated, obtaining 0.7 cm mycelium discs that were placed inside 9 cm Petri dishes with 10 mL of PDA. After the mycelium was inoculated in the agar, micro-drops of 20 μL of EEP at 1, 1.5, 2, and 3% *w*/*v* were added, as a control, without propolis. The test was carried out in triplicate. The Petri dishes were closed and incubated at 20 °C for five days and the radial growth of the microorganism was evaluated.

### 2.4. Preparation of the Antifungal Film

The films were elaborated following the casting methodology, using the steps described by Abarca et al. (2022) [29] with minor modifications. The solution was prepared with 4% *w*/*v* of sodium alginate and 4% *w*/*v* of glycerol as plasticizer in distilled water and was stirred on a heating plate at 70 °C and 500 rpm for 2 h. Once the mixture was homogenized, the EEP was added at 5 or 10% *w*/*v* and the mixture was stirred again for 30 min at 40 °C and 700 rpm. Finally, 70 g of the prepared solution was poured into glass Petri dishes, which were dried for 48 h at 45 °C.

### 2.5. Film Characterization

#### 2.5.1. Thermogravimetric Analysis (TGA)

To evaluate the thermal stability of the antifungal films, a TGA was performed by subjecting the samples to a sustained temperature increase. This test was carried out on a TGA/DSC 1 STARe System, Mettler Toledo, Greinfensee, Switzerland. The films were subjected to a thermal sweep from 12 °C to 900 °C with a heating rate of 10 °C/min, with a flow (50 mL/min) of high purity nitrogen [30].

#### 2.5.2. Color

The films were evaluated by colorimetric analysis using a Hunterlab Mini Scan XE Plus 45/0-L colorimeter, which used the CIELab system to measure the brightness parameter (L*) and the yellow-blue (b*) and red-green (a*) color parameters. Five measurements were made at different points of each film containing the antimicrobial agent, using as a control a film without the agent. The color difference (ΔE) was calculated with respect to the control film, using the expression ΔE = [(ΔL*)^2^ + (Δa*)^2^ + (Δb*)^2^]^1/2^ [29].

#### 2.5.3. Opacity Index

To evaluate the variation in the opacity produced by the incorporation of the antifungal agent to the polymeric matrix (alginate), film samples of 1 × 4.5 cm were cut and introduced into the reading area of a UV-Vis Spectronic^®^ Genesys 5 spectrophotometer, where the absorbance of the samples was determined at 650 nm [31]. The opacity index was calculated using the following equation:opacity = Absorbance (650 nm)/Thickness

#### 2.5.4. Film Thickness

To determine the influence of the antifungal agent on the polymer matrix, the thickness of the films was measured with a Mitutoyo Digimatic digital micrometer (model ID-C112, Kawasaki, Japan). The measurements were carried out at 15 different points, to obtain an average value of the thickness plus a standard deviation [28].

#### 2.5.5. Film Morphology

To evaluate the effect of EEP on the sodium alginate film, an analysis was performed by scanning electron microscopy (SEM), using films without the agent as control. The samples were analyzed in a CrossBeam (FIB-SEM), AURIGA Compact. The analyzes were carried out with a voltage of 10 kv and a magnification of 2000× (Abarca et al., 2017).

#### 2.5.6. Fourier Transform Infrared Spectroscopy (FTIR)

The functional groups in the different films were analyzed by FTIR on an IR spectrometer, Bruker, model Alpha. Spectra were obtained in the range 4000 to 400 cm^−1^ [32].

### 2.6. Antifungal Activity

#### 2.6.1. In Vitro Evaluation of the Antifungal Activity of the Film against Filamentous Molds

The antifungal activity of the films was evaluated with respect to the in vitro mycelial growth of strains isolated from cheese storage warehouses. Each sample was analyzed in triplicate. The method used was carried out according to Solano et al. (2020) [33] with some modifications, as follows: about 10 mL of PDA were poured into a Petri dish. Once the agar gelled, a 0.7 cm agar disk was inoculated in the center with a mycelium of the isolated strain, obtained from the initial culture of the mold. From the films, 0.9 cm diameter samples were cut with sterile scissors, disinfected with UV light and placed in the Petri dish around the mycelium to evaluate the behavior of the mold when in contact with the film containing the antifungal compound. All this procedure was carried out in a laminar flow hood. Finally, the plates were incubated for 5 days at 20 °C, conditions considered optimal for fungi growth. Films without the antifungal agent were used as controls. Once the incubation time was completed, the radial growth of the mycelium was measured and compared with the radial growth of the mold incubated with the control films.

#### 2.6.2. Evaluation of the Antifungal Activity of the Film/Coating against Filamentous Molds in Pieces of Cheese

A solution of sodium alginate with EEP was prepared according to Section 2.4, but without the film formation step. The dip coating procedure was performed as described by Tabassum & Khan, (2020) [34] with modifications. Once the mixture was made, fresh and ripened cheese was cut into 5 × 5 cm pieces. The coating was applied in two ways: immersing the pieces of fresh and ripened cheese in a container with a homogeneous mixture of the coating for 30 s; or painting the surface of the pieces of fresh and ripened cheese with a brush, covering their entire surface. Subsequently, the samples were left at room temperature (12 °C) for 30 days with a relative humidity of 85% for further evaluation. This stage was carried out in a preliminary manner, which will be extended in a new study, analyzing sensory and technological parameters in the product.

### 2.7. Statistical Analysis

A multiple rank analysis was proposed to study the effects of incorporation of propolis into alginate-based films. The variable studied was propolis concentration, ranging between 1% to 3% *w*/*v*. As response, this study includes optical properties (L*, a*, b*, ∆E* and opacity) and thickness as variables measured. The number of samples was n = 5 and n = 15 for optical parameters and thickness, respectively. The analysis of total data was carried out by Statgraphics Centurion XVII software by one-way ANOVA and statistical differences were found by LSD with 95% confidence level. 

## 3. Results and Discussion

### 3.1. Concentration of Total Phenolic Compounds in the Ethanolic Extract of Propolis

According to Ramón-Sierra et al. (2019) [19] phenolic compounds are a diverse group of secondary metabolites produced by multiple plant species and represent the most diverse class of compounds present in propolis.

The percentage of total polyphenols was 20.60%, and the concentration of total polyphenols expressed in GAE was 20,595.37 mg GAE/100 g. This result agrees with that obtained by Zhang et al. (2016) [35] who found 19,280 mg GAE/100 g of phenolic compounds in ethanol extract of Chinese propolis (EECP), 21,770 mg GAE/100 g in ethanol extract of *Eucalyptus* propolis (EEEP), and 13,507 mgGAE/100 g in ethanol extract of *Baccharis* propolis (EEBGP) [35]. The content of phenolic compounds in propolis is highly variable because it depends on the diversity of plant species to which bees have access during its preparation [36]. On the other hand, according to Bakchiche, (2017) [37] each species of bee has a habit of collecting resin that increases the regional and interspecific variations in the composition of propolis.

### 3.2. Minimum Inhibitory Concentration (MIC)

To determine the MIC, different concentrations of EEP (1% (b), 1.5% (c), 2% (d) and 3% (e) *w*/*v*) were analyzed in vitro in direct contact with the fungal mycelium on PDA plates and a control sample without EEP (a) (Figure 1). The tests were incubated for 5 days at 20 °C. After this period, it was possible to observe radial growth inside the Petri dishes at concentrations of 1 and 1.5% *w/v* of EEP (Figure 1b,c). One of the advantages of this technique, for which it has been described as the “gold standard” compared to other methods that assess susceptibility to antimicrobials, is that, in addition to confirming unusual resistances, it gives definitive answers when the result obtained by other methods is indeterminate [38]. In this case, the MIC was defined as the lowest concentration of an antimicrobial agent that visibly inhibited the growth of a microorganism after an incubation period in contact with it [30]. The first concentration that stopped the radial growth of the filamentous molds studied was 2% *w*/*v* (Figure 1d). In general, this fungal behavior depends on the species under study, since there are fungal organisms that have physiological and biochemical versatility in the presence of a stressor, which in this case was EEP. A physiological defense mechanism to counter stress factors is the increase in growth rate or increased biochemical activity translated into the production of lytic enzymes and reactive oxygen species. The antimicrobial effect of propolis depends on several factors, such as the type of microorganism (e.g., bacteria, yeast, virus, mold), cell concentration (with a high inoculum, larger amounts of propolis were required for antimicrobial action), and the mode of action (a growth retardation rather than complete inhibition) [39]. Propolis, in the form of EEP, has been used successfully in in vitro inhibition studies of several phytopathogenic fungi, such as *Alternaria alternata* and *Fusarium oxysporum f.* sp. *melonis*, among others, that cause food spoilage [40].

Knowing the MIC value of the EEP studied, against filamentous molds that contaminate and alter the manufacture of ripened cheeses, allows us to know the necessary amount to be incorporated into a coating system to achieve a reasonable extension of the shelf life of this type of food product.

### 3.3. Thermogravimetric Analysis (TGA)

Thermal stability is important for the materials used in food packaging applications. TGA is used to evaluate the thermal stability of polymers, and to monitor the impact of heating on the loss of weight of the volatilized compounds and the degradation of the different compounds that are added [17,41,42]. The analyzes were carried out with the sodium alginate film (control) and the films with different concentrations of EEP.

The TGA curve (Figure 2a) shows that all films had three stages of weight loss. The first stage was observed at around 100 °C, with a 5% mass loss due to the loss of water [30,43]. The second loss occurred in the range of 200–320 °C and was attributed to the decomposition of glycerol compounds and carbohydrate polymers in the film matrix [44].

Weight loss stages can be better observed in the curves of the derivative of the TGA (Figure 2b), which coincides with the findings of Bagheri et al. (2019) [17] who observed that the progressive weight loss at 201 °C in alginate films is mainly due to glycerol evaporation. They further point out that the maximum weight loss near temperatures of 207, 209 and 212 °C, with termination temperatures at around 230 °C, were observed for the films dried at 90, 57 and 25 °C, respectively.

Films with 10% EEP had higher degradation temperatures. In general, the films with EEP exhibited a higher thermal stability than the control films, mainly due to the incorporation of 5 and 10% EEP. Control film, 5% EEP and 10% EEP reached temperatures of 267.8, 271.6 and 314 °C, respectively, by losing 70% of their mass.

Salama et al. (2018) [32] obtained similar results and reported that the degradation of sodium alginate from 174 °C to 495 °C is due to the breakdown of glycosidic bonds, the evolution of carbon dioxide and the destruction of the hydroxyl group, and even to the decomposition of the plasticizer (glycerol).

The weight loss curves continued until the third stage at 400–500 °C onwards, which, according to Wai Chun et al. (2021) [45] is attributed to loss of soot by combustion or pyrolysis in this temperature range. 

### 3.4. Color and Opacity

Opacity and color are important parameters to consider in food packaging, as they influence the acceptance and commercial success of the product [28]. The film surface color was evaluated by measuring the values of L, a* and b* (L = 0 (black) to 100 (white); a* = (−) green to (+) red; b* = (−) blue to (+) yellow), using the CIELab scale [15]. The total color difference (∆E) of the films is shown in Table 1, where there is a difference in the ∆E value with respect to the control (*p* < 0.05), obtaining values over 0.4, which surpasses human perception. The luminosity values (L) decreased, while the chromatic values a* and b* increased in direct proportion as EEP concentration increased from 5 to 10% (*p* < 0.05). These color values were affected by the incorporation of EEP, a result similar to that of Khodayari et al. (2019) [46] who when adding EEP to a polymeric matrix of lactic polyacid, observed that the color parameters were different from those of the control [46]. Increasing EEP concentration resulted in deeper, yellow-colored films compared to clear colored alginate control films. This is a consequence of the colored substances present in propolis [20]. This point is relevant when choosing food packaging material for foods such as cheese, as opacity is generally not desired, because it makes it difficult for the potential consumer to correctly visualize the condition of the food [47]. However, in photosensitive foods opacity is desirable as it provides protection against deterioration produced by light [48]. In this specific case, despite the opacity parameter being increased, the coloring has the same tendency as the color of the food to be packaged, such as cheese, thus tending to yellow. Film opacity with the antifungal agent was significantly higher (*p* < 0.05) than in the control films (Table 1). This could be because the addition of EEP improves the reaction with sodium alginate, making the film structure more dense and less transparent. Films composed of an aromatic amino acid containing benzene exhibit some blocking effect towards UV light. Therefore, the addition of EEP could have a similar effect [49]. According to Abarca et al. (2017) [28] and Pastor et al. (2010) [50] the changes in opacity can be attributed to the agglomeration of the agents added to the polymeric matrix, which generates a reduction in the passage of light through the surface (difference in the refractive index) that favors the dispersion of the light.

Cazón, Vázquez and Velazquez, (2018) [51] pointed out that the opacity or transparency of a film could also be affected by thickness. In the present study, this is supported by the information from Table 2, which shows the statistically significant differences between the thickness of the control film and the 10% EEP film.

### 3.5. Thickness

The range of thicknesses for films with 5 and 10% EEP were in a range of 0.33 to 0.39 mm, respectively. In the case of the control film, the average was 0.31 mm (Table 2).

There were no significant differences between control films and 5% EEP films (*p* > 0.05), but 10% EEP films did have a significant increase in thickness with respect to the others (*p* < 0.05).

Shahrampour et al. (2020) [52] obtained similar results for thickness when adding pectin to sodium alginate films. Increasing the pectin content increases the thickness of composite films and is associated with the unique colloidal properties of the compound, including thickening, suspension and interaction between components. On the other hand, Mahcene et al. (2020) [15] observed that thickness varies depending on the type of essential oil incorporated into the sodium alginate films; and Correa-Pacheco et al. (2019) [14], add that a film must have a thickness of at least 0.3 mm to prevent the loss of moisture in a food.

According to Liu et al. (2016) [53], film thickness is related to the type of film components and their interaction. In addition, the final thickness is an important parameter that affects the opacity, the water vapor permeability and the mechanical properties of films; therefore, this parameter depends on the preparation method and the drying conditions [15,52].

### 3.6. Morphology

In general, micrographs of the surface of the control films (Figure 3a) showed a homogeneous structure (without insoluble particles). In contrast, micrographs of EEP films (Figure 3b,c) showed a rougher surface with aggregated particles. This same result was reported by Pastor et al. (2020) [50], who added EEP to a polymeric matrix of hydroxy-propyl-methyl-cellulose.

Regarding the control film, Bagheri et al. (2019) [17] report that sodium alginate films dried above 57 °C show a structure with few pores and cracks, because solutions formed at higher temperatures have a lower viscosity, allowing the escape of microscopic air bubbles. Bodini et al. (2013) [54] reported that the increase in EEP concentrations in a gelatin polymeric matrix produces an increase in porosity due to the distribution of the agent, causing the structure to be irregular compared to the control film, consistent with what was observed in this study.

### 3.7. Fourier Transform Infrared Spectroscopy (FTIR)

FTIR spectroscopy was used to characterize the intermolecular interaction between the film matrix and the phenolic compounds of the EEP, measuring the transmittance in the range of 4000–500 cm^−1^.

The FTIR spectra of the control films and the 5 and 10% EEP films are shown in Figure 4. All films showed a similar pattern of FTIR spectrum, with most of the absorption bands showing characteristics of alginate films, but with different transmittance intensities.

The control film showed bands at 3423 cm^−1^ for the O-H stretching vibration, at 2929 cm^−1^ for the C-H stretching, between 1645 and 1452 cm^−1^ for the asymmetric and symmetric stretching of -COO-, and a band of approximately 1080 cm^−1^, which was assigned to a mannuronic unit [18,32,41].

All films exhibited a pronounced first absorption band in the range of 3600–3000 cm^−1^, which indicated the presence of OH functional groups [30,32]. The absorbed water molecules are due to the presence of glycerol as a plasticizer, which is characterized by being hygroscopic [55]. The lack of absorption bands in the spectrum between 2600 and 1800 cm^−1^ indicated that there is no presence of functional groups with triple bonds, as mentioned by Asma et al. (2014) [55].

For EEP films, the bands observed at 1700 cm^−1^ are attributable to the stretching of C=O within the carboxylic group. Bands at 1600 cm^−1^ correlate with the stretching of C=C within the aromatic ring [56], which indicates functional groups of phenolic compounds of propolis. The stretching of C=O, -C=C- C=O, -C=C- (within the aromatic ring), CC- (within the aromatic ring), CO (esters, ethers) and CO (polyols) found in phenolic components were observed at 1800–1000 cm^−1^ of the FTIR spectra [14,20].

The FTIR results suggest that the profiles of the prepared films are similar, due to the homogenization of propolis and glycerol with sodium alginate. In addition, it is known that there is intermolecular interaction between the hydroxyl and carboxyl groups of the sodium alginate matrix and the functional groups of phenolic compounds of EEP [14].

### 3.8. Evaluation of the Antifungal Activity of the Films against Filamentous Fungi and Ripened Cheese

The addition of EEP to the sodium alginate film had effects at the two concentrations studied (5 and 10% *w*/*v*). Prepared sodium alginate films with EEP were 0.34 mm thick. Figure 5 shows that the control film had a radial growth of filamentous mold, while the 5 and 10% EEP films had no radial growth. No zones of inhibition were observed, but when the fungal microorganism came into direct contact with the active film, its growth stopped. Pastor et al. (2010) [50] observed a similar effect with *Penicillium italicum* strains and hydroxy-propyl-methyl-cellulose films with propolis, as they did not observe antifungal effects. However, growth data indicated that composite films were effective in reducing microbial growth and that effectiveness increased with increasing propolis concentration, although complete inhibition of fungal growth was not possible for *Penicillium italicum*. On the other hand, antifungal effects were seen for *Aspergillus niger* strains at higher concentrations of propolis.

Cortés-Higadera et al. (2019) [57] demonstrated a notable decrease in the growth of *Aspergillus flavus* using combinations of chitosan nanoparticles and propolis nanoparticles at 30% and 40%, which indicated a synergistic effect between the components. Regarding the antifungal activity of propolis, according to Mattiuz et al. (2015) [23] treatments with 2.5% propolis completely inhibited radial growth of *Colletotrichum gloeosporioides*, always showing 0 mm for mycelial diameter. Propolis has been shown to have activity against various fungi [23,57,58]; however, in this study, EEP had no antifungal activity but did have fungistatic activity, stopping the growth and development of the fungus. This is because propolis has antioxidant molecules that are fungistatic, in which the components galangin and pinocembrin stand out due to the high concentrations of polyphenols [21].

According to Sforcin, 2016 [59] pinocembrin shows activity against *Penicillium italicum*, as it stops the growth of the mycelium by acting on its respiration and energy homeostasis, which leads to the rupture of the cell membrane.

Propolis has flavonoid compounds, phenolic acids, and esters, quinones, coumarins, sesquiterpenes and steroids, which confer antibiotic properties [23]. The antioxidant properties of propolis are due to phenolic compounds that donate hydrogen ions to free radicals to protect the cell from oxidation reactions, and propolis can scavenge free radicals, which are a main cause of lipid, nucleic acid, and protein oxidation [21].

After physically and chemically characterizing the films and studying their antifungal activity in vitro, tests were performed on pieces of cheese. Here, the film was applied as a coating on pieces of ripened and fresh cheese by dipping or painting (Figure 6). The cheeses were stored for 30 days at room temperature to observe the possible fungal growth on the surface.

At the end of the storage time, the ripened cheese pieces had a dark and dry appearance compared to the fresh cheese. On the other hand, no fungi were observed, neither on the surface of the fresh cheese nor on the surface of the ripened cheese.

The application of the coating on the pieces of cheese demonstrated that shelf life can be extended by incorporating the active agent into the polymer matrix. Despite being a preliminary test, it gives an idea of its application and the effect on the food matrix. On the other hand, active compounds are being applied in polymeric matrices to increase fruit preservation. Solano et al. (2020) [33] used lauric acid as a natural antifungal agent against the growth of *Colletotrichum tamarilloi* in tomato, showing that the active compound delayed fungal growth by two weeks for the yellow variety and one week for the red variety.

## 4. Conclusions

An active film, using sodium alginate as the polymeric matrix and EEP as the active agent, was produced using the casting method. The characterization tests indicated that the incorporation of EEP significantly affected the color and thickness of the film. In addition, EEP provided greater thermal stability against weight loss. Regarding the functionality as an active film, a fungistatic mode of action was observed in in vitro tests, as fungal growth stopped when coming into direct contact with EEP films.

The test on cheese showed that the active mixture applied to the food matrix (cheese) as a film (by immersion or brush painting) was able to prevent the growth of filamentous fungi in a period of 30 days of storage at room temperature.

On the other hand, the modification of the properties of the active film could be attributed to the interactions of the polymeric matrix, the polyphenols and the components of propolis, as verified in the FTIR analysis. The developed film has the potential to be used in the food industry as an antimicrobial packaging material.

Results from, this study can be of use for those processing plants looking for alternative strategies to mitigate dairy product contamination from microorganisms. Further studies should also focus on other dairy products and how to combat biofilm caused by microorganisms within the industry, thus avoiding contamination in the process lines. Finally, caution must be paid, as the chemical properties of propolis can be different depending on its location of production, and further research is advised to compare propolis from different locations. Future efforts should focus on the effects of EEP on sensory characteristics in the food matrix and the evaluation of the shelf life on cheeses and other dairy products.

## Figures and Tables

**Figure 1 foods-12-01633-f001:**
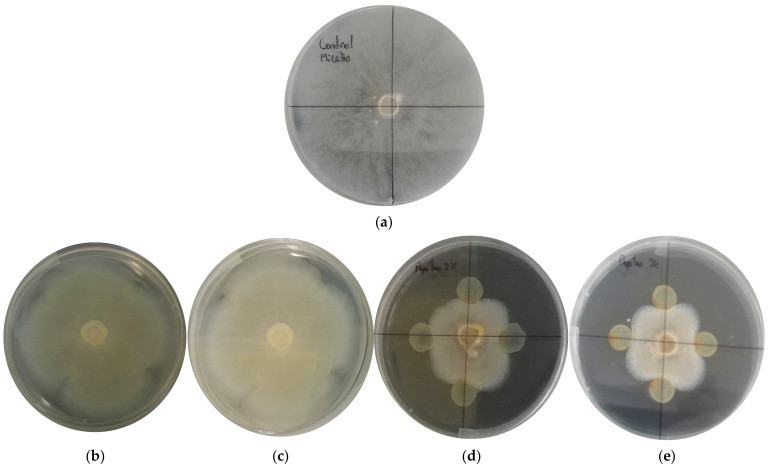
Minimum inhibitory concentration (**a**) control sample; (**b**) 1% *w*/*v* ethanolic extract of propolis (EEP), (**c**) 1.5% *w*/*v* ethanolic extract of propolis (EEP); (**d**) 2% *w*/*v* ethanolic extract of propolis (EEP); (**e**) 3% *w*/*v* ethanolic extract of propolis (EEP).

**Figure 2 foods-12-01633-f002:**
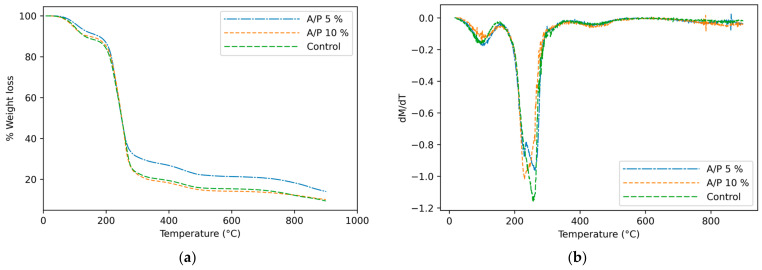
Thermogravimetric analysis of the control film (Alginate) and the EEP films (A/P) (**a**). Derivatives of thermal degradation curves of the films (**b**).

**Figure 3 foods-12-01633-f003:**
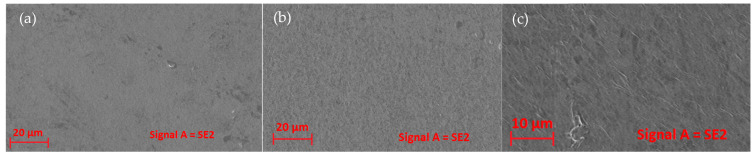
Micrographs of film surfaces; control film (**a**), alginate film with 5% EEP (**b**) and alginate film with 10% EEP (**c**).

**Figure 4 foods-12-01633-f004:**
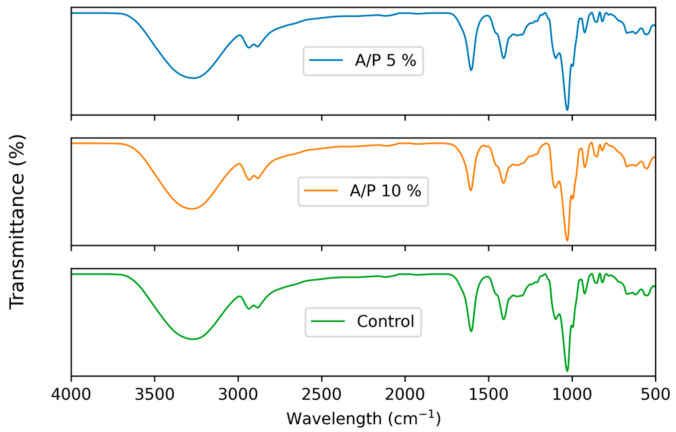
FTIR spectra of alginate films containing ethanolic extract of propolis (EEP).

**Figure 5 foods-12-01633-f005:**
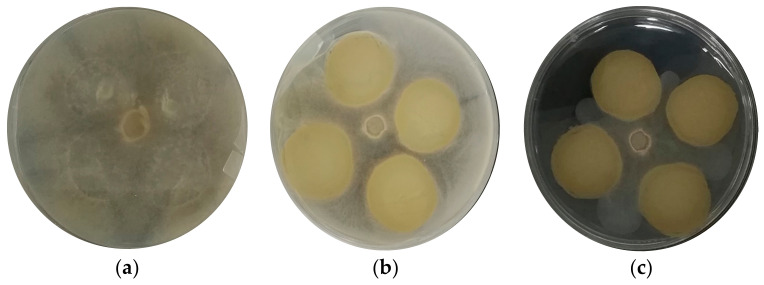
Antifungal activity of films. (**a**) Control film; (**b**) 5% EEP film; (**c**) 10% EEP film.

**Figure 6 foods-12-01633-f006:**
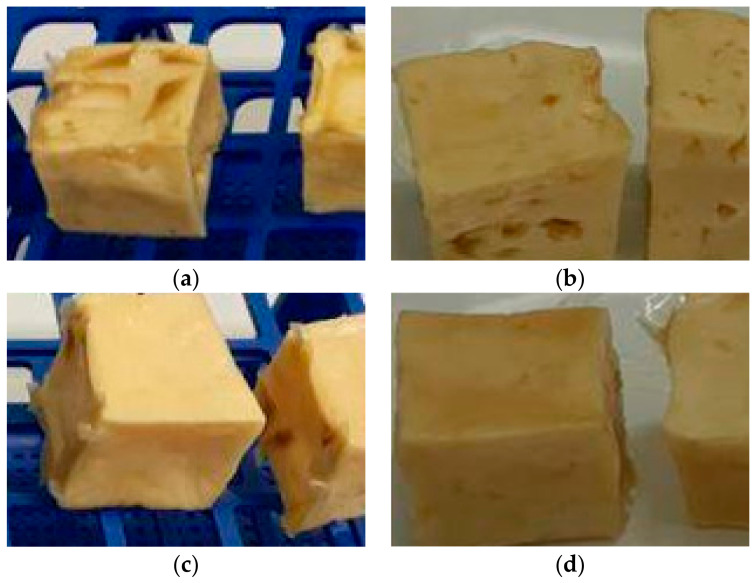
Antifungal activity on samples of ripened and fresh cheese. (**a**) Immersion coated ripened cheese. (**b**) Ripened cheese covered by brush painting. (**c**) Immersion coated fresh cheese. (**d**) Fresh cheese covered by brush painting.

**Table 1 foods-12-01633-t001:** Optical properties of the prepared films and their opacity.

Treatment	a*	b*	L*	∆E	Opacity
Control film	−2.13 ± 0.03 ^a^	6.30 ± 0.03 ^a^	84.94 ± 0.04 ^a^	-	2.66 ^a^
5% EEP film	3.56 ± 0.02 ^b^	30.14 ± 0.03 ^b^	62. 27 ± 0.05 ^b^	69.27 ± 0.05 ^a^	4.43 ^b^
10% EEP film	8 ± 0.2 ^c^	28.52 ± 0.04 ^c^	55.56 ± 0.03 ^c^	62.96 ± 0.03 ^b^	4.19 ^c^

Each value represents an average of five replicates with its corresponding standard deviation. Data were statistically analyzed with the multiple range test using Fisher’s least significant difference (LSD) method. L* refers to the lightness coordinate, and its value ranges from 0 for perfect black to 100 for perfect white; a* and b* are chromaticity coordinates on the green-red (−a* = green; +a* = red) and blue-yellow (−b* = blue; +b* = yellow) axes. The different letters indicate significant differences in each column (*p* < 0.05).

**Table 2 foods-12-01633-t002:** Thickness of the films.

Samples	Thickness (mm)
Control film	0.31 ± 0.043 ^a^
5% EEP film	0.33 ± 0.061 ^a^
10% EEP film	0.39 ± 0.038 ^b^

The different letters indicate significant differences in each column (*p* < 0.05). Each value represents an average of 15 replicates with its corresponding standard deviation. Data were statistically analyzed with the multiple range test using Fisher’s least significant difference (LSD) method. The different letters indicate significant differences (*p* < 0.05).

## Data Availability

The datasets generated and analyzed in the current study are available from the corresponding author upon reasonable request.

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
