# Peer review of "Development and Characterization of Films with Propolis to Inhibit Mold Contamination in the Dairy Industry"

_foods, 2023, doi:10.3390/foods12081633_

Round 1
Reviewer 1 Report
Abarca et al. tested the physical properties of an alginate film enriched with propolis extract. In addition, they performed antifungal/fungistatic in vitro tests and covered cheese pieces with propolis extract coating.
The abstract does not rigorously follow the structure of introduction, methods, results, discussion and conclusion. It describes methods and results. We do not know from the abstract why the research was conducted and what its impact is.
The introduction provides an explanation of current challenges related to food spoilage and the need for better methods to prevent these problems. The background information provided is relevant, but the research objectives are not clearly based on the sentence intended to support them: "Due to its antibacterial, antiviral, anti-inflammatory, and anesthetic properties, propolis is considered a functional compound in food packaging [25]." None of these properties is relevant if the authors intend to study the antifungal effect of propolis. They need to choose a reference that demonstrates the antifungal property of propolis.
The authors have provided clear and concise descriptions of the materials and methods used in their study. They have also included details on the extraction of the antifungal agent, determination of total polyphenols, minimum inhibitory concentration analysis, antifungal film preparation, film characterization, film morphology, Fourier transform infrared spectroscopy, antifungal activity, and statistical analysis. Whereas, the authors used a single-source propolis sample. How can they conclude that propolis, in general, is useful for the purpose?
Using the term "in vivo" is not correct when the authors test the antifungal effect of propolis as a coating agent on cheese, because cheese cannot be an in vivo subject. In vivo studies involve tests on living organisms such as humans, animals or plants.
Lines 403-405: The authors claim that propolis is fungistatic because it is an antioxidant. Do they mean that antioxidant molecules are fungistatic?
It is not clear what happened to the control cheese pieces. If the cheese dried and darkened during the 30-day incubation period, exsication is suspected. In dried cheese, mold cannot grow. Dehydration itself is a form of preservation without using propolis extract. What was the humidity during storage? What was the difference between treated and control?
Propolis can alter the organoleptic and textural values of cheese. Unfortunately, these issues were not addressed in this study.
3. Results and discussion followed by 5. Conclusions.
Overall, the manuscript needs to be improved, in particular as indicated above.
Author Response
Response to Reviewer 1 Comments
The article < Development and characterization of films with propolis to in-hibit mold contamination in the dairy industry
Abarca et al. tested the physical properties of an alginate film enriched with propolis extract. In addition, they performed antifungal/fungistatic in vitro tests and covered cheese pieces with propolis extract coating.
The abstract does not rigorously follow the structure of introduction, methods, results, discussion and conclusion. It describes methods and results. We do not know from the abstract why the research was conducted and what its impact is.
AUTHORS: This was revised as suggested. See abstract
The introduction provides an explanation of current challenges related to food spoilage and the need for better methods to prevent these problems. The background information provided is relevant, but the research objectives are not clearly based on the sentence intended to support them: "Due to its antibacterial, antiviral, anti-inflammatory, and anesthetic properties, propolis is considered a functional compound in food packaging [25]." None of these properties is relevant if the authors intend to study the antifungal effect of propolis. They need to choose a reference that demonstrates the antifungal property of propolis.
AUTHORS: This was revised as suggested. See lines 88-89
The authors have provided clear and concise descriptions of the materials and methods used in their study. They have also included details on the extraction of the antifungal agent, determination of total polyphenols, minimum inhibitory concentration analysis, antifungal film preparation, film characterization, film morphology, Fourier transform infrared spectroscopy, antifungal activity, and statistical analysis. Whereas, the authors used a single-source propolis sample. How can they conclude that propolis, in general, is useful for the purpose?
AUTHORS: a short note was made at the end of the conclusion. See lines 495-497
Using the term "in vivo" is not correct when the authors test the antifungal effect of propolis as a coating agent on cheese, because cheese cannot be an in vivo subject. In vivo studies involve tests on living organisms such as humans, animals or plants.
AUTHORS: This was revised as suggested and changes were made accordingly in the text
Lines 403-405: The authors claim that propolis is fungistatic because it is an antioxidant. Do they mean that antioxidant molecules are fungistatic?
AUTHORS: This sentence has been revised as suggested. See lines 441-442
It is not clear what happened to the control cheese pieces. If the cheese dried and darkened during the 30-day incubation period, exsication is suspected. In dried cheese, mold cannot grow. Dehydration itself is a form of preservation without using propolis extract. What was the humidity during storage? What was the difference between treated and control?
AUTHORS: For clarity, this was inserted: Cheese pieces were preserved at 12°C and 85% relative humidity. Three different concentrations of EEP were applied to the cheese pieces at 5 and 10% w/v. The control had no EEP and all treatments were stored under the same conditions. See lines 106-109
Propolis can alter the organoleptic and textural values of cheese. Unfortunately, these issues were not addressed in this study.
AUTHORS: this is an interesting point but unfortunatly was not analyzed. We have inserted a note at the end of the concilusion as a further step to do. See lines 497-498
- Results and discussion followed by 5. Conclusions.
AUTHORS: This was changed, see line 477
Overall, the manuscript needs to be improved, in particular as indicated above.
AUTHORS: All suggested changes were done accordingly

Reviewer 2 Report
The manuscript can be improved in terms of the writing style and quality of presentation, i.e. Figure 1 - state what in vitro analysis, Figure 6 - include antifungal measurements, Figure 7 - require more clear and consistent presentation of the samples.
The discussion section needs to be improved, i.e. Line 277 no discussion given.
Line 455, inappropriate conclusion.
Author Response
Response to Reviewer 2 Comments
The article < Development and characterization of films with propolis to in-hibit mold contamination in the dairy industry
The manuscript can be improved in terms of the writing style and quality of presentation, i.e. Figure 1 - state what in vitro analysis, Figure 6 - include antifungal measurements, Figure 7 - require more clear and consistent presentation of the samples.
AUTHORS: This was revised as suggested. See figures
The discussion section needs to be improved, i.e. Line 277 no discussion given.
AUTHORS: This was revised as suggested. See discussion Lines 291-293
Line 455, inappropriate conclusion.
AUTHORS: This was revised as suggested. Was eliminated

Reviewer 3 Report
The authors have submitted a manuscript in which they evaluate the potential bioactive films to inhibit mold contamination of ripened cheeses. The application of these films is interesting and could have a potential interest in the food sector. Otherwise, I consider that the manuscript should undergo a deep review process to be published, in order to complete and improve the exposition of the results. The work is based on the potential application of a bioactive film to the dairy industry, but there are not enough results to support this ambitious hypothesis. I would recommend completing the study of the shelf life of this product (cheese) with more analyses. Another option could be to reorient the objective of this article concerning the obtaining and characterization of these films. I encourage authors to make the needed changes to publish this work.

Author Response
Response to Reviewer 3 Comments
The article Development and characterization of films with propolis to in-hibit mold contamination in the dairy industry
The authors have submitted a manuscript in which they evaluate the potential bioactive films to inhibit mold contamination of ripened cheeses. The application of these films is interesting and could have a potential interest in the food sector. Otherwise, I consider that the manuscript should undergo a deep review process to be published, in order to complete and improve the exposition of the results. The work is based on the potential application of a bioactive film to the dairy industry, but there are not enough results to support this ambitious hypothesis. I would recommend completing the study of the shelf life of this product (cheese) with more analyses. Another option could be to reorient the objective of this article concerning the obtaining and characterization of these films. I encourage authors to make the needed changes to publish this work.
AUTHORS: this study is a proof-of-concept where EEP was tested on a dairy food matrix. At this point, the nature of the study is exploratory. However, we tried to use all methods available in our laboratory to asses the potential of EEP to avoid proliferation of spoilage microorganisms. We also consider that further research efforts should focus on shelf life, sensory characteritics and the use of propolis from different locations. These ideas were inserted at the end of the conclusion section.
Perhaps we can request the Editorial to change this study from ARTICLE to COMMUNICATION so results and intepretation does not seem to be ambitious as refered by the reviewer.
Please let us know what route should be taken.

Round 2
Reviewer 1 Report
The authors have revised the manuscript in line with the comments, but the language and the smoothness of the text could be improved.
"Data Availability Statement: Not applicable."
Are the raw data not available?
Author Response
The article Development and characterization of films with propolis to inhibit mold contamination in the dairy industry
The authors have revised the manuscript in line with the comments, but the language and the smoothness of the text could be improved.
AUTHORS: this has been reviseed as suggested
"Data Availability Statement: Not applicable."
Are the raw data not available?
AUTHORS: this was changed to Data Availability Statement: The datasets generated and analyzed in the current study are available from the corresponding author upon reasonable request.

Reviewer 3 Report
Dear authors,
I would ask you to introduce and consider the suggested changes to proceed with the publication of the article.
I have reviewed the article and the author's response. Only some of the questions have been answered (some of these changes appear directly in the text). The authors have also made some changes to the manuscript that improve its overall quality. Considering the author's response (they are wondering whether it would be necessary to pass the article to communication), I do not know if they have not made these changes because they are waiting for an answer.
Author Response
The article Development and characterization of films with propolis to inhibit mold contamination in the dairy industry
I would ask you to introduce and consider the suggested changes to proceed with the publication of the article. I have reviewed the article and the author's response. Only some of the questions have been answered (some of these changes appear directly in the text).
AUTHORS: all changes have been done accordingly
See lines 29-30
See lines 90-97
See lines 105-107
See lines 210
See lines 491-493
The authors have also made some changes to the manuscript that improve its overall quality. Considering the author's response (they are wondering whether it would be necessary to pass the article to communication), I do not know if they have not made these changes because they are waiting for an answer.
AUTHORS: We have revised with the Foods Editorial and based on the contents of this manuscript, we have an Article full research paper.
We have made some notes to clarify the objective of the paper and improve the rationale of it. See lines 90-97. Additionally, a last comment was added in the conclusion section to reinforce the facta that sensory chartcateristics and shelf life should be done in future studies. At the moment we do not have more samples for extra analysis and we have performed all analysis needed to reach our scientifc objetive.
